# Two-Period Study Results from a Large Italian Hospital Laboratory Attesting SARS-CoV-2 Variant PCR Assay Evolution

Flora Marzia Liotti,[a] Flavio De Maio,[a] Chiara Ippoliti,[a] Giulia Santarelli,[b] Francesca Romana Monzo,[a] Michela Sali,[a,b] Rosaria Santangelo,[a,b] Francesca Ceccherini-Silberstein,[c] Maurizio Sanguinetti,[a,b] Brunella Posteraro[b,d]

[a]Dipartimento di Scienze di Laboratorio e Infettivologiche, Fondazione Policlinico Universitario A. Gemelli IRCCS, Rome, Italy

[b]Dipartimento di Scienze Biotecnologiche di Base, Cliniche Intensivologiche e Perioperatorie, Università Cattolica del Sacro Cuore, Rome, Italy

[c]Dipartimento di Medicina Sperimentale, Università di Roma Tor Vergata, Rome, Italy

[d]Dipartimento di Scienze Mediche e Chirurgiche, Fondazione Policlinico Universitario A. Gemelli IRCCS, Rome, Italy

Flora Marzia Liotti and Flavio De Maio contributed equally to this article and are co-first authors. Author order was determined randomly.

**ABSTRACT** In keeping with the evolution of severe acute respiratory syndrome coronavirus 2 (SARS-CoV-2), the COVID-19 causative agent, PCR assays have been developed to rapidly detect SARS-CoV-2 variants, which have emerged since the first (Alpha) variant was identified. Based on specific assortment of SARS-CoV-2 spike-protein mutations (ΔH69/V70, E484K, N501Y, W152C, L452R, K417N, and K417T) among the major variants known to date, Seegene Allplex SARS-CoV-2 Variants I and Variants II assays have been available since a few months before the last (Omicron) variant became predominant. Using S gene next-generation sequencing (NGS) as the SARS-CoV-2 variant identification reference method, we assessed the results of SARS-CoV-2-positive nasopharyngeal swab samples from two testing periods, before ($n = 288$, using only Variants I) and after ($n = 77$, using both Variants I and Variants II) the appearance of Omicron. The Variants I assay allowed correct identification for Alpha (37/37), Beta/Gamma (28/30), or Delta (220/221) variant-positive samples. The combination of the Variants I and Variants II assays allowed correct identification for 61/77 Omicron variant-positive samples. While 16 samples had the K417N mutation undetected with the Variants II assay, 74/77 samples had both ΔH69/V70 and N501Y mutations detected with the Variants I assay. If considering only the results by the Variants I assay, 6 (2 Beta variant positive, 1 Delta variant positive, and 3 Omicron variant positive) of 365 samples tested in total provided incorrect identification. We showed that the Variants I assay alone might be more suitable than both the Variants I and Variants II assays to identify currently circulating SARS-CoV-2 variants. Inclusion of additional variant-specific mutations should be expected in the development of future assays.

**IMPORTANCE** Omicron variants of SARS-CoV-2 pose more important public health concerns than the previously circulating Alpha or Delta variants, particularly regarding the efficacy of anti-SARS-CoV-2 vaccines and therapeutics. Precise identification of these variants highly requires performant PCR-based assays that allow us to reduce the reliance on NGS-based assays, which remain the reference method in this topic. While the current epidemiological SARS-CoV-2 pandemic context suggests that PCR assays such as the Seegene Variants II may be dispensable, we took advantage of NGS data obtained in this study to show that the array of SARS-CoV-2 spike protein mutations in the Seegene Variants II assay may be suboptimal. This reinforces the concept that initially developed PCR assays for SARS-CoV-2 variant detection could be no longer helpful if the SARS-CoV-2 pandemic evolves to newly emerging variants.

**KEYWORDS** SARS-CoV-2, spike protein, variant detection, PCR assay, amino acid mutation, next-generation sequencing

Address correspondence to Maurizio Sanguinetti, maurizio.sanguinetti@unicatt.it.

The authors declare no conflict of interest.

Since their marketing in early 2020 (1), real-time reverse transcriptase PCR (rRT-PCR)-based COVID-19 diagnostic assays (hereafter referred to as PCR assays), and the interpretation of their results, have had to adapt to the continuous evolution of severe acute respiratory syndrome coronavirus 2 (SARS-CoV-2) (2–4), which is notorious as the COVID-19 pandemic causative agent (5). Viral evolution has been decorated with variants of concern (VOCs), which became apparent as the SARS-CoV-2 genome began to accumulate an array of mutations, predominantly in the spike (S) protein-encoding gene (6). Accordingly, S gene mutation patterns have allowed the World Health Organization (WHO) to classify SARS-CoV-2 VOCs as Alpha, Beta, Gamma, Delta, and Omicron (7, 8). These variants are also known as B.1.1.7, B.1.351, P1, B.1.617.2, and B.1.1.529, respectively, according to the phylogenetic assignment of named global outbreak (Pango) lineages. Unlike the WHO, the U.S. Centers for Disease and Control and Prevention (CDC) has also classified the Epsilon (Pango lineage, B.1.427/B.1.429) SARS-CoV-2 variant as a VOC (2). Currently, both the U.S. CDC and WHO consider the lastly emergent Omicron SARS-CoV-2 variant as the only circulating VOC worldwide (https://www.cdc.gov/coronavirus/2019-ncov/variants/variant-classifications.html; https://www.who.int/activities/tracking-SARS-CoV-2-variants).

While the efficacy of vaccines and monoclonal antibodies might be heavily undermined by Omicron (9, 10), it is clear that, relative to earlier SARS-CoV-2 variants, the "heavily mutated" Omicron variant (11) makes its detection by S gene targeting PCR assays more challenging (3, 4). Notable amino acid mutations on the SARS-CoV-2 S protein include the H69/V70 deletion (ΔH69/V70) in Alpha B.1.1.7 and Omicron B.1.1.529; N501Y in Alpha B.1.1.7, Beta B.1.351, Gamma P1, and Omicron B.1.1.529; E484K in Beta B.1.351 and Gamma P1; K417N in Beta B.1.351 and Omicron B.1.1.529; and two others (K417T and L452R) in Gamma P1 and Delta B.1.617.2 (6). The ΔH69/V70 mutation is not present in the BA.2, a sister lineage of the originally designated BA.1, where BA is an alias for B.1.1.529 (12). The mutation L452R was found in Epsilon B.1.427/B.1.429 together with three other mutations, including the W152C (6).

In view of their clinical importance, rapid detection of SARS-CoV-2 variants has become affordable through the development of PCR assays capable of detecting multiple mutations, including the S protein mutations mentioned above (13). In this context, based on the S mutation assortment among SARS-CoV-2 variants, Seegene Inc. (Seoul, South Korea) has recently developed two multiplex PCR assays, namely, the Allplex SARS-CoV-2 Variants I assay (https://www.seegene.com/assays/allplex_sars-cov-2_variants_i_assay) and Allplex SARS-CoV-2 Variants II assay (https://www.seegene.com/assays/allplex_sars-cov-2_variants_ii_assay). Of these assays (hereafter referred to as Seegene Variants I and Seegene Variants II, respectively), one was marketed in February 2021 and the other one in April 2021, almost concurrently with the global spread of the Delta variant that outcompeted other VOCs as of May 2021 (3, 4). This occurred 6 months before the Omicron's global spread (12). If used concertedly, these assays could allow important SARS-CoV-2 variant(s) to be detected, but studies on their performance have, expectedly, been restricted to non-Omicron variants (13–15).

Here, we report on the SARS-CoV-2 variant detection results of nasopharyngeal swab (NPS) samples with a PCR-based SARS-CoV-2 diagnosis that were tested during the May 2021 through November 2021 COVID-19 pandemic period (before Omicron's appearance; 288 samples with only the Seegene Variants I assay) or during the December 2021 through January 2022 COVID-19 pandemic period (after Omicron's appearance; 77 samples with both the Seegene Variants I and II assays), respectively. Results were compared with those obtained by S gene next-generation sequencing (NGS) on each sample, and agreement rates were assessed in the two study periods.

(This work was presented in part at the 32nd European Congress of Clinical Microbiology and Infectious Diseases [ECCMID] held in Lisbon, Portugal [23 to 26 April 2022].)

## RESULTS

We included 365 SARS-CoV-2-positive NPS samples, of which 283 (77.5%) had a diagnostic PCR nucleocapsid (N) gene cycle threshold ($C_T$) value of ≤25. Based on the S gene and/or whole-genome NGS analysis, 221 (76.7%) of 288 samples from the first study period were positive for Delta (B.1.617.2), 37 (12.9%) for Alpha (B.1.1.7), 28 (9.7%) for Gamma (P.1), and 2

(A)

| | Amino acid mutations detected (+) or not detected (−) in the spike protein from the indicated SARS-CoV-2 variant by | | | | | | |
| --- | --- | --- | --- | --- | --- | --- | --- |
| | Seegene Variants I assay | | | Seegene Variants II assay | | | |
| Variant | ΔH69/V70 | E484K | N501Y | W152C | L452R | K417N | K417T |
| Alpha (B.1.1.7) | + | − | + | − | − | − | − |
| Beta (B.1.351) | − | + | + | − | − | + | − |
| Gamma (P.1) | − | + | + | − | − | − | + |
| Delta (B.1.617.2) | − | − | − | − | + | − | − |
| Epsilon (B.1.427/B.1.429) | − | − | − | + | + | − | − |
| Omicron (B.1.1.529) | + | − | + | − | − | + | − |

(B)

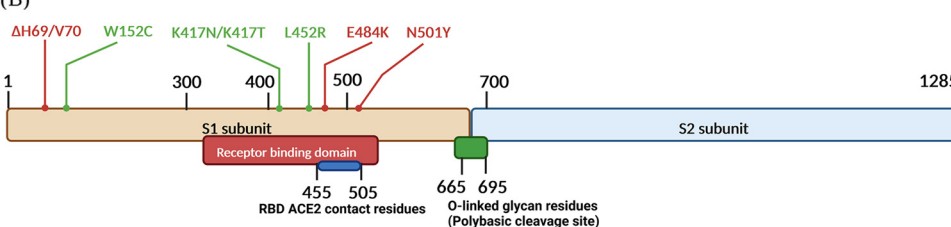

**FIG 1** SARS-CoV-2 VOC or non-VOC identification scheme by PCR assays (A) based on the detection of seven amino acid mutations in the SARS-CoV-2 spike (S) protein (B). The Seegene Allplex Variants I assay (here named the Seegene Variants I assay) detects the SARS-CoV-2 RdRP gene in addition to the indicated mutations. The Seegene Allplex Variants II assay (here named the Seegene Variants II assay) detects four mutations, including W152C, which is specific for the Epsilon variant (which is not a VOC according to the WHO). The S protein mutation position is shown, with E484K and N501Y involving RBD ACE2 contact amino acid residues.

(0.7%) for Beta (B.1.351). Seventy-seven (100%) of 77 samples from the second study period were positive for Omicron (B.1.1.529).

We interpreted Seegene Variants I assay results based on the S mutation identification scheme shown in Fig. 1, which includes the most frequent or characteristic mutations across SARS-CoV-2 variants. Accordingly (Table 1), 37 (100%) of 37 Alpha variant-positive samples, 28 (100%) of 28 Gamma variant-positive samples, and 220 (99.6%) of 221 Delta variant-positive samples were correctly identified. Overall, results were incorrect for 3 samples, providing a 99.0% (285/288 samples) agreement rate between the Seegene Variants I assay and NGS results. One Delta variant-positive sample was misidentified as Beta/Gamma variant positive, and both Beta variant-positive samples were not identified.

According to the S mutation identification scheme shown in Fig. 1, Seegene Variants I used in combination with the Seegene Variants II assay allowed correct identification for 61 of 77 Omicron variant-positive samples (Table 1). This resulted in an agreement rate between the Seegene Variants I and Variants II assay and NGS results of 79.2% (61/77 samples). Of 16 samples with apparently incorrect results, 13 had only the ΔH69/V70 and N501Y mutations detected and 3 had no mutation detected. In the 3 samples, the diagnostic $C_T$ value (mean ± standard deviation [SD])—which strongly correlates with the sample's viral load—did not differ from that of the remaining 13 samples (32.0 ± 2.9 and 30.6 ± 3.3, respectively; $P > 0.05$). Interestingly, 74 (96.1%) of 77 samples had both ΔH69/V70 and N501Y mutations detected with the Seegene Variants I assay, and 61 of 74 samples had the K417N mutation detected with the Seegene Variants II assay (Table 1). In the 61 samples, compared to the diagnostic $C_T$ value (18.4 ± 4.9), $C_T$ values for the ΔH69/V70, N501Y, or K417N detection were 20.0 ± 4.2, 23.5 ± 4.1, and 33.1 ± 3.7, respectively (mean ± SD). If considering only the results by the Seegene Variants I assay, 3 (3.9%) of 77 samples indeed had results that did not agree with NGS results. To clarify this issue, we stratified the Seegene Variants I and Variants II assay results of 77 samples by a diagnostic PCR $C_T$ value of ≤25 or >25 (Table 2). Among the samples with a $C_T$ value of >25, the proportion of samples that tested Omicron variant positive with the Seegene Variants I assay was significantly higher than with both the Seegene Variants I and Variants II assays ($P < 0.001$). Therefore, samples with higher $C_T$ values were more likely to have no K417N mutation detected and, consequently, an S mutation pattern not consistent

**TABLE 1** Assessment of SARS-CoV-2 variant detection results compared with next-generation sequencing results for SARS-CoV-2 positive NPS samples

| Samples (no.) stratified by SARS-CoV-2 variant with indicated diagnostic PCR $C_T$ value[a] | | S mutations (no.) detected with: | | Variant identified (no. of samples interpreted as variant positive/no. of samples tested)[b] | Percent agreement between Seegene assay(s) and NGS (95% confidence interval)[c] |
|---|---|---|---|---|---|
| | | Seegene Variants I assay | Seegene Variants I and II assays | | |
| Alpha (37) | $19.5 \pm 3.4$ | ΔH69/V70 (37), N501Y (37) | | 37/37 | 100.0 (90.6–100.0) |
| Beta (2) | $16.5 \pm 3.5$ | E484K (1) | | 0/2 | – |
| Gamma (28) | $20.9 \pm 3.2$ | E484K (28), N501Y (28) | | 28/28 | 100.0 (87.9–100.0) |
| Delta (221) | $21.2 \pm 5.6$ | E484K (1), N501Y (1) | | 220/221 | 99.6 (97.5–99.9) |
| Total (288) | $20.9 \pm 5.2$ | – | | 285/288 | 99.0 (97.0–99.7) |
| Omicron (77) | $21.0 \pm 6.9$ | | ΔH69/V70 (74), N501Y (74), K417N (61) | 61/77 | 79.2 (68.9–86.8) |

[a]NPS samples included in the study (n = 365) had a previous SARS-CoV-2-positive result with a diagnostic PCR assay (i.e., Seegene Allplex SARS-CoV-2 assay) that provided a $C_T$ value of ≤40 for all assay targets (i.e., E, RdRP/S, and N SARS-CoV-2 genes). The indicated $C_T$ value represents the mean ± standard deviation of $C_T$ values obtained for each single sample within the variant group. The lowest of diagnostic PCR $C_T$ values (e.g., N gene $C_T$) obtained for each sample was used for calculation.

[b]Seegene Variants I assay results (n = 288) or Seegene Variants I and Variants II assay results (n = 77) were interpreted as "variant identified" according to the S mutation-based identification scheme illustrated in Fig. 1. The Beta variant-positive samples were interpreted as "no variant identified" because of partially (i.e., presence of only one of variant-defining mutations) or fully (i.e., absence of variant-defining mutations) incomplete S mutation pattern, respectively. In 1 of 221 Delta variant-positive samples, E484K and N501Y mutations were incorrectly detected (i.e., consistent with a Beta/Gamma S mutation pattern). Of 77 Omicron variant-positive samples, 74 had an Omicron-consistent S mutation pattern with the Seegene Variants I assay, and 61 of 74 had an Omicron-consistent S mutation pattern when also tested with the Seegene Variants II assay.

[c]NGS results consisted of S gene sequencing results and were used as a comparator for the agreement rate calculation with respect to the SARS-CoV-2 variant identification. The symbol "–" indicates that a value was not calculated or reported. NGS, next-generation sequencing.

with the presence of Omicron variant when both Seegene assays were used (Table 2). However, it is possible that in these samples, a PCR dropout due to a low viral load (i.e., a low PCR target) had occurred rather than concomitant mutations had precluded the detection of the interrogated mutation.

Figure 2 provides an overview of amino acid mutations (i.e., substitutions or deletions) detected by the S gene NGS in groups of samples according to the Alpha (37 samples), Beta (2 samples), Gamma (28 samples), Delta (221 samples), or Omicron (77 samples) SARS-CoV-2 variant identified. Mutations included amino acid substitutions or deletions known to define or not define SARS-CoV-2 variants (6). Regarding Alpha/Omicron (ΔH69/V70 and N501Y) or Beta/Gamma (E484K and N501Y) variant-defining mutations detectable by the Seegene Variants I assay (Fig. 1), NGS analysis did not detect either E484K or N501Y in one Beta variant-positive sample and N501Y in 38 Omicron variant-positive samples. Regarding Beta (K417N), Gamma (K417T), Delta (L452R), Epsilon (W152C and L452R), or Omicron (K417N) variant-defining mutations detectable by the Seegene Variants II assay (Fig. 1), NGS analysis did not detect K417N in 1 Beta variant-positive sample or in 41 Omicron variant-positive

**TABLE 2** SARS-CoV-2 variant detection results for 77 Omicron variant-positive NPS samples stratified by diagnostic PCR $C_T$ values[a]

| $C_T$ value used for stratification (no. of samples in each group) | No. of samples detected as Omicron variant positive/ no. of samples tested by the indicated assay(s) (percentage) | | P value[b] |
|---|---|---|---|
| | Only Seegene Variants I | Both Seegene Variants I and II | |
| ≤25 (55) | 55/55 (100) | 55/55 (100) | NA |
| >25 (22) | 19/22 (86.4)[c] | 6/22 (27.3)[d] | <0.001 |
| Total (77) | 74/77 (96.1) | 61/77 (79.2) | 0.008 |

[a]The indicated $C_T$ values are the values obtained by testing samples with a diagnostic SARS-CoV-2 PCR assay (i.e., Seegene Allplex SARS-CoV-2 assay) and refer to the lowest $C_T$ values for the N (73 samples), RdRP/S (0 samples), and E (4 samples) SARS-CoV-2 genes targeted by the assay.

[b]The chi-square test was used to assess the statistically significant difference between percentages (i.e., P value of <0.05). This was not available (NA) in one of three instances.

[c]Three samples with a negative assay result (i.e., with neither ΔH69/V70 nor N501Y detected) had a $C_T$ value (mean ± SD) of 32.0 ± 2.9.

[d]Thirteen samples with a negative assay result (i.e., with either ΔH69/V70 or N501Y but no K417N detected) had a $C_T$ value (mean ± SD) of 30.6 ± 3.3.

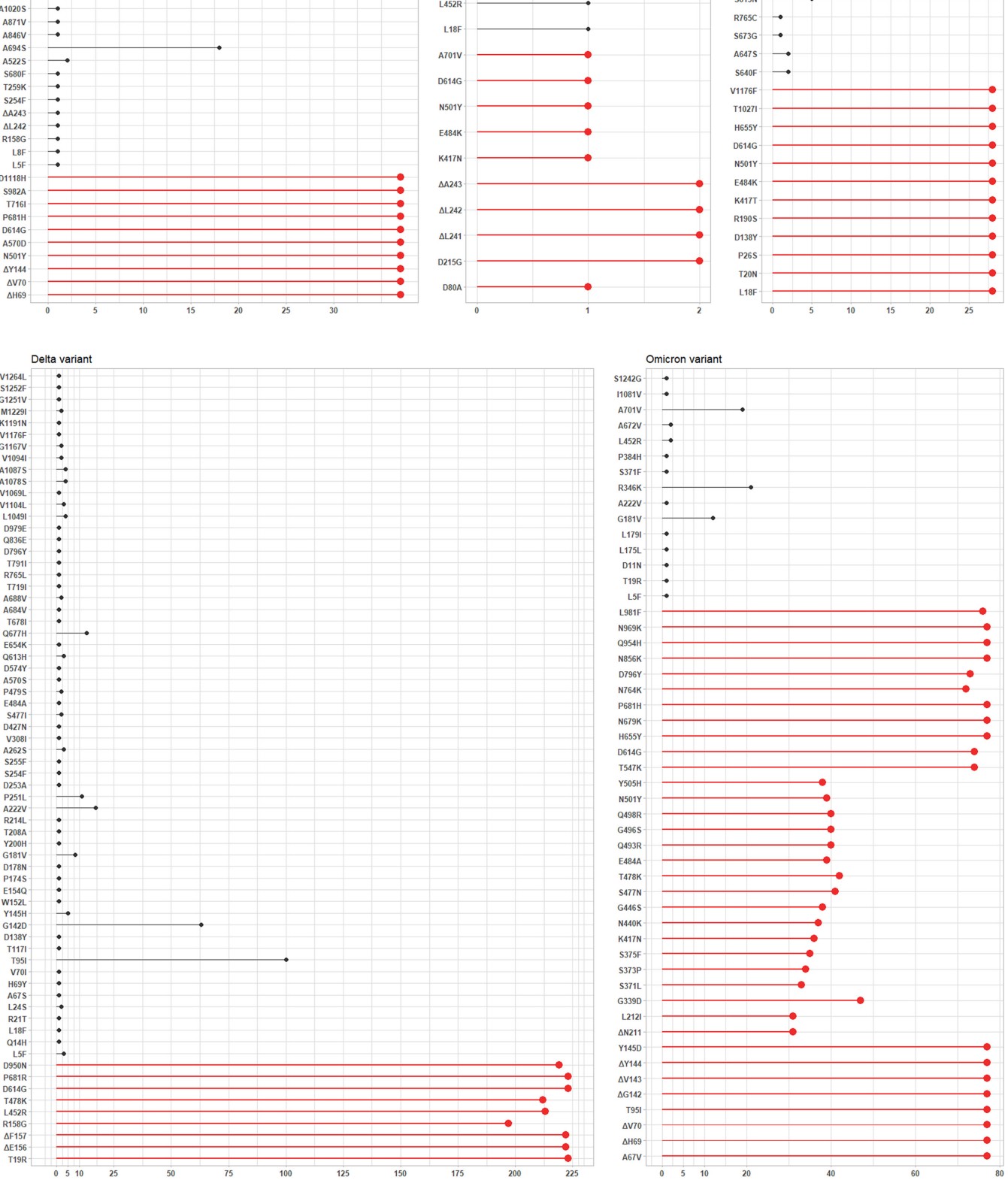

**FIG 2** Next-generation sequencing results for SARS-CoV-2-positive NPS samples showing amino acid mutations in the SARS-CoV-2 spike (S) gene-encoded protein. Sample results are shown according to the type of SARS-CoV-2 variant (Alpha, Beta, Gamma, Delta, or Omicron) identified. In each graph, colors allow grouping of S protein mutations according to whether (red) or not (black) they define the indicated SARS-CoV-2 variant.

samples or L452R in 8 Delta variant-positive samples. Unsurprisingly, the W152C mutation was not detected in any, except for one (Delta variant positive), of the study samples. Overall, NGS analysis detected 95 additional SARS-CoV-2 variant-nondefining mutations, with A694S, S813N, T95I, and R346K being predominant in Alpha (18/37 samples), Gamma (5/28 samples), Delta (100/221 samples), and Omicron (21/77 samples) variant-positive samples, respectively.

## DISCUSSION

Our evaluation of Seegene Variants I and Variants II assays on SARS-CoV-2-positive NPS samples using the S gene NGS as a comparator showed excellent agreement rates for Alpha/Omicron, Beta/Gamma, or Delta/Epsilon SARS-CoV-2 variant identification results with the Seegene Variants I assay. Six samples (two Beta variant positive, one Delta variant positive, and three Omicron variant positive) provided incorrect identification results. Conversely, the agreement rate was acceptable for Omicron variant identification results with both the Seegene Variants I and Variants II assays—this regarded the second study period, when only SARS-CoV-2 Omicron variant-positive samples were tested. Sixteen Omicron variant-positive samples provided incorrect identification results.

Like S mutation-targeting PCR assays described elsewhere (16–18), Seegene SARS-CoV-2 PCR assay evolution embraced a cluster of clinically relevant SARS-CoV-2 S gene mutations (seven mutations, if using both the Seegene Variants I and Variants II assays). We used an interpretive algorithm to categorize five major (Alpha, Beta, Gamma, Delta, and Omicron) SARS-CoV-2 VOCs sequentially identified since September 2020, which are known to harbor (and share with each other) multiple S gene mutations compared to the SARS-CoV-2 Wuhan-Hu-1 isolate's reference sequence (as reviewed in references 4, 7, and 19). At least 1 (and up to 15 in the Omicron variant) of these mutations are in the S protein receptor-binding domain (RBD), which binds the host angiotensin-converting enzyme 2 (ACE2) and is the target of many neutralizing antibodies (8, 19). Two RBD mutations, namely, N501Y and E484K, are associated with increased ACE2 affinity (and virus transmissibility) and thought to compensate for the attenuated ACE2 affinity owing to the presence of K417N/T mutations (20), which might indeed act as immune/antibody escape mutations. The L452R mutation within the RBD seems to affect both virus transmissibility and immune/antibody escape. Finally, ΔH69/V70 mutations are associated with increased infectivity and, importantly, an S gene target failure (SGTF) in some multiplex PCR assays (4). Looking at the Seegene Variants I and Variants II assay results by targeted S mutations revealed that the Variants I assay allowed detection of ΔH69/V70 mutations in all but 3 of 114 (37/37 Alpha and 74/77 Omicron) variant-positive samples (6). According to our NGS results, sequences from all Omicron variant-positive samples showed ΔH69/V70 mutations and, consistently, none of these samples was positive for the Omicron BA.2, which lacks ΔH69/V70 mutations (6). We also found that the Variants II assay allowed detection of the K417N mutation, which was included alongside K417T in the assay to discriminate *a priori* between Beta and Gamma variants and *a posteriori* between Alpha and Omicron variants (Fig. 1), in all but 16 of 77 Omicron variant-positive samples.

Discrepant results between SARS-CoV-2 variant detection PCR assays and S gene NGS, which remains the gold standard to confirm SARS-CoV-2 infection with a specific variant (21), may be attributed to low viral loads in the samples tested (13, 18). Consistent with a diagnostic PCR N gene $C_T$ value of $\leq 25$ in $\sim 80\%$ of samples, all SARS-CoV-2 sequences in our study reached 90% of S gene coverage (data not shown). This ensured good-quality samples to assess the Seegene Variants I and Variants II assays as well as reliable NGS data to assess samples for the presence of S gene mutations, including those detectable by the Seegene Variants I and Variants II assays. As expected, while allowing SARS-CoV-2 variant determination, the S gene NGS assay used in this study (see Materials and Methods for details) showed diminished coverage (i.e., amplicon dropout) particularly in SARS-CoV-2 Omicron variant-positive samples. This resulted in 53% (41/77) of Omicron variant sequences with the K417N mutation. Consistently, recent (unpublished) assessment results of SARS-CoV-2 Omicron variant sequences from the GISAID database (https://www.gisaid.org/) showed that the prevalences of the

K417N mutation were 57% in BA.1 and 94% in BA.2, whereas the prevalences of the N501Y mutation were 86% in BA.1 and 92% in BA.2. These findings hampered us from corroborating the apparent failure of the Seegene Variants II assay to detect the K417N mutation in ~21% of Omicron variant-positive samples. Otherwise, in the samples with the K417N mutation undetected, $C_T$ diagnostic values of >25 seemed to predict successful mutation detection by the Seegene Variants II assay, suggesting that the combined use with the Seegene Variants I assay may be restricted to the samples with a high viral load.

At the time of submission of the manuscript, new SARS-CoV-2 VOCs Pango lineages such as Omicron BA.4 and BA.5 have appeared, thus displacing the earlier Omicron BA.1 (B.1.1.529.1) and BA.2 (B.1.1.529.2). While considering our study particularly relevant to current Omicron variant-dominated contexts, we were unable to compare Omicron variant-positive sample results with those from previous studies (i.e., performed before Omicron variants' appearance). In a study published in early 2022 (13), the Seegene Variants II assay showed 100% sensitivity compared to NGS for (Alpha, Beta, Gamma, and Delta) SARS-CoV-2 variant identification in 72 NPS samples. Almost concurrently, two independent studies evaluated the Seegene Variants I assay in comparison with NGS on 408 and 115 NPS samples (14, 15), showing 100% sensitivity for ΔH69/V70, N501Y, and E484K detection. Excluding Omicron variant-positive samples, sensitivity (i.e., percent agreement with NGS) of the Seegene Variants I assay in our study was slightly below the above value.

We are aware the limitation of interpreting Seegene Variants I and Variants II assay results with prior knowledge of the circulating SARS-CoV-2 variant(s)/lineage(s) in the study periods. To be clinically useful, these assays should allow for precise discrimination between variants (i.e., Alpha and Omicron, Beta and Gamma, or Delta and Epsilon), which is not allowed with the Seegene Variants I assay alone and, instead, could be with both the Seegene Variants I and Variants II assays. Despite relying on simplified protocols, NGS analysis is labor-intensive and mainly employed in reference clinical microbiology laboratories. We are also aware of the (inevitable until now) limitation of using research-use-only (RUO) versions of commercial SARS-CoV-2 S gene sequencing assays on the samples included in the study. This might have biased the Seegene Variants I and Variants II assay evaluation analysis performed by us.

In conclusion, our experience with the Seegene Variants I and Variants II assays during the mid-2021/early 2022 COVID-19 pandemic confirms the good performance of these assays and, in the meantime, emphasizes the need for continuous evolution in the SARS-CoV-2 variant detection PCR assays. We believe that currently Seegene Variants I may offer a rapid and simple way to identify SARS-CoV-2 Omicron variants, while the Seegene Variants II assay or a potential development thereof (i.e., to include additional variant-specific S gene mutations) may be unnecessary until the epidemiological COVID-19 pandemic context remains unmodified.

## MATERIALS AND METHODS

**Study design and clinical samples.** This retrospective study was carried out at and approved by the Institutional Ethics Committee (reference no. 0018900/21) of the Fondazione Policlinico Universitario A. Gemelli IRCCS hospital of Rome, Italy. We included NPS samples that had tested positive with the Seegene Allplex SARS-CoV-2 assay (https://www.seegene.com/assays/allplex_sars_cov_2_assay), i.e., that had a diagnostic cycle $C_T$ of ≤40 for all the assay's targets, such as the envelope (E), RNA-dependent RNA polymerase (RdRP)/S, and nucleocapsid (N) SARS-CoV-2 genes. Two hundred eighty-three samples had a $C_T$ value of ≤25 (range, 9.0 to 24.9), whereas the remaining 82 samples had a $C_T$ value of >25 (range, 25.1 to 35.6). The N gene $C_T$ value was ≤25 in 283 samples, and the E gene $C_T$ value was ≤25 in 273 samples. Thus, 82 samples had N gene $C_T$ values slightly exceeding the cutoff values that have been suggested to optimize the sequencing success (22, 23). Residual aliquots from NPS samples, i.e., SARS-CoV-2 RNA extract-containing samples, were kept at −20°C until either NGS or variant detection analysis (see below for details). Samples were collected in two study periods (from May 2021 to November 2021 and from December 2021 to January 2022) to account for 365 samples included in total. While we adopted a random selection criterion for sample inclusion in this study, we could include only 77 samples in the second period due to the study budget's constraints. As these samples were obtained in a period completely dominated by the Omicron variant (see below for identification details), the sample size in the second period was somewhat comparable with that of single non-Omicron variants (see below for identification details) in the first period. We had also studied part of these samples previously (24).

**SARS-CoV-2 S gene sequencing.** Samples from both the study periods were subjected to NGS analysis of the SARS-CoV-2 S protein-encoding gene. We used the CleanPlex SARS-CoV-2 panel (Paragon Genomics, Hayward, CA) or the SARS-CoV-2 S gene kit (Arrows Diagnostics, Genoa, Italy) for samples ($n = 67$ and $n = 64$, respectively) collected before 1 July 2021 and the COVIDSeq assay kit (Illumina, San Diego, CA) for samples ($n = 234$) collected after 1 July 2021. This was the date of implementing the whole-genome sequencing (WGS)-based SARS-CoV-2 variant surveillance program in the Lazio Region in Italy (24), which is our study's location. For the groups of 67 and 234 samples mentioned above, WGS data were also available (see Table S1 in the supplemental material). Briefly, purified and quantified amplicons from each SARS-CoV-2 RNA extract-containing sample were used to prepare a DNA library, which was paired-end sequenced on a MiSeq instrument (Illumina) according to the manufacturer's recommendations. Sequencing reads were analyzed using the SOPHiA GENETICS platform for COVID-19, Covid Analyst software (SmartSeq s.r.l., Novara, Italy), or Illumina DRAGEN COVIDSeq Pipeline software as appropriate, and their quality was checked using NextStrain (https://nextstrain.org/).

Based on WGS data, samples were identified as positive for the Alpha B.1.1.7 ($n = 13$), Gamma P1 ($n = 2$), Delta B.1.617.2 ($n = 209$), or Omicron B.1.1.529 ($n = 77$) variant according to the Pango lineage designation. Based on S protein mutation patterns, 63 of 64 remaining samples were identified as Alpha ($n = 24$), Beta ($n = 1$), Gamma ($n = 26$), or Delta ($n = 12$) variant positive according to the WHO designation. Another sample was identified as Beta positive based on the presence of S protein deletions from positions 241 to 243 and D215G substitution, which have been found exclusively in the SARS-CoV-2 Beta variant (25).

According to technical note specifications for using the Illumina COVIDSeq assay kit, some amplicons were expected to show diminished coverage in SARS-CoV-2 variants, including Omicron (26). The S gene regions spanning from nucleotides 22038 to 23122 (22038 to 22262, 22346 to 22516, 22650 to 22797, and 22903 to 23122), which include the K417N mutation, had median coverage values (interquartile ranges [IQR]) of 8 (3 to 14), 25 (6.3 to 41), 16.5 (8 to 32.3), and 4.5 (1 to 12), respectively. These values were lower than those of other remaining regions included in the analysis (data not shown). However, percentages (median value [IQR]) of non-N bases and coverage of $\geq 30\times$ for SARS-CoV-2 whole-genome sequences from the study samples were 98.9 (96.3 to 99.7) and 96.4 (94.6 to 98.5), respectively.

**SARS-CoV-2 variant detection testing.** Samples from the first study period ($n = 288$) were subjected to the Seegene Variants I (for proper assay designation, see above), which is designed to detect the SARS-CoV-2 RdRP gene and three ($\Delta$H69/V70, E484K, and N501Y) S mutations. Samples from the second study period ($n = 77$) were subjected to the Seegene Variants I assay as well as the Seegene Variants II assay (for proper assay designation, see above), which is designed to detect four (W152C, L452R, K417N, and K417T) S mutations. The assays were simultaneously run and were performed according to the manufacturer's instructions. These assays are multiplex real-time PCR assays to allow for qualitative detection and differentiation of SARS-CoV-2 variant-consistent S amino acid mutations in a single tube. An internal control is included and processed with each sample to monitor the integrity of the results. The limit of detection of the Seegene Variants I assay is 698 copies/mL, as determined using spiked NPS samples with serially diluted synthetic DNA of the RdRP gene and the $\Delta$H69/V70, E484K, and N501Y mutation-harboring S gene mentioned above (14). For each assay, the Seegene Viewer automatically analyzes the results and provides the $C_T$ value for each of the four targets and the internal control by means of five separate fluorescence-based detection channels. Figure 1 shows the identification scheme for WHO-designated VOC (Alpha, Beta, Gamma, Delta, and Omicron) or non-VOC (Epsilon) SARS-CoV-2 variants, which relies on the presence or absence of detection of the SARS-CoV-2 S mutations mentioned above. This scheme was used to interpret assay results as variant positive or negative for all the samples included in the study. A positive RdRP gene result was required to assess as reliable any S mutation detection result by the assay(s). As expected, we found no negative results for RdRP gene detection. The $C_T$ value relative to each mutation's detection by the assay(s) was also considered for assay result assessment.

**Statistical analysis.** Study results were reported as numbers with percentages or as means $\pm$ SD, as appropriate. Differences between *a priori* established groups were assessed using the chi-square test or the Student $t$ test, as appropriate. Percent agreement values, with their respective confidence intervals (CIs), were calculated for result comparisons between the Seegene Variants I assay, used alone or in combination with the Seegene Variants II assay, and the NGS reference method. Additionally, sensitivity, specificity, positive predictive value (PPV), and negative predictive value (NPV) were calculated per Seegene assay variant detections (see Table S2 in the supplemental material). Statistical analysis was performed using Stata 15 (StataCorp, College Station, TX) or GraphPad Prism 7 (GraphPad Software, San Diego, CA) software, and a $P$ value of $<0.05$ was considered statistically significant.

**Data availability.** Whole-genome sequences in FASTQ format were uploaded on the GISAID database (https://www.gisaid.org/) through the IRIDA (Integrated Rapid Infectious Disease Analysis) ARIES (Advanced Research Infrastructure for Experimentation in GenomicS) platform (https://irida.iss.it/irida21-aries/login), whereas S gene sequences obtained as described above were uploaded on the Sequence Read Archive (SRA) repository under accession number PRJNA880628.

## SUPPLEMENTAL MATERIAL

Supplemental material is available online only.
**SUPPLEMENTAL FILE 1**, PDF file, 0.2 MB.

## ACKNOWLEDGMENTS

We thank Seegene Inc. (Seoul, South Korea) for providing a financial support that allowed us to acquire assay kits for SARS-CoV-2 variant detection and sequencing in this study, and we thank the European Community under HORIZON-HLTH-2021-CORONA-01 for

funding the EuCARE project "European cohorts of patients and schools to advance response to epidemics" (grant agreement number 101046016).

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
