## [Reviewer comments · Microbiology Spectrum]

Two-Period Study Results from a Large Italian Hospital Laboratory Attesting SARS-CoV-2 Variant Detection PCR Assays' Evolution

Flora Liotti, Flavio De Maio, Chiara Ippoliti, Giulia Santarelli, Francesca Monzo, Michela Sali, Rosaria Santangelo, Francesca Ceccherini-Silberstein, Maurizio Sanguinetti, and Brunella Posteraro

Corresponding Author(s): Maurizio Sanguinetti, Fondazione Policlinico Universitario

Review Timeline:

Submission Date:	July 27, 2022
Editorial Decision:	September 6, 2022
Revision Received:	October 5, 2022
Accepted:	October 24, 2022

Editor: Heba Mostafa

Reviewer(s): Disclosure of reviewer identity is with reference to reviewer comments included in decision letter(s). The following individuals involved in review of your submission have agreed to reveal their identity: Ryan John Dikdan (Reviewer #1); Raghda Eldesouki (Reviewer #2)

Transaction Report:

DOI: <https://doi.org/10.1128/spectrum.02922-22>

September 6, 2022

Prof. Maurizio Sanguinetti
Fondazione Policlinico Universitario
Microbiology
L.go A. Gemelli 8
Rome, RM 168
Italy

Re: Spectrum02922-22 (Two-Period Study Results from a Large Italian Hospital Laboratory Attesting SARS-CoV-2 Variant Detection PCR Assays' Evolution)

Dear Prof. Maurizio Sanguinetti:

Link Not Available

Sincerely,

Heba Mostafa

Journals Department
Reviewer comments:

Reviewer #1 (Comments for the Author):

See attached file.

Reviewer #2 (Comments for the Author):

Authors need to address the sensitivity, specificity, positive predictive values of the assays.

Staff Comments:

Preparing Revision Guidelines

Please return the manuscript within 60 days; if you cannot complete the modification within this time period, please contact me. If you do not wish to modify the manuscript and prefer to submit it to another journal, please notify me of your decision immediately so that the manuscript may be formally withdrawn from consideration by Microbiology Spectrum.

VOC have changed quite a lot over time, so this paper should reflect the most recent status, which is that the US CDC only considers Omicron and its subvariants as VOC <https://www.cdc.gov/coronavirus/2019-ncov/variants/variant-classifications.html> , whereas WHO states that Omicron is the only currently circulating VOC and Alpha, Beta, Gamma, and Delta are the previously circulating VOC: <https://www.who.int/activities/tracking-SARS-CoV-2-variants/>

While the Pango lineages are more specific for mutational patterns, in this context the WHO designated VOC names of Alpha, Beta, etc. should be used when talking about the virus mutations, if no further breakdown of lineage is being discussed. Lines 68-72

Around lines 75, the current state of genotyping methods in the field of SARS-CoV2 variants should be discussed in the introduction for proper context of the reader. Ref 15 is a good start.

When you use d69-70 and N501Y to identify Alpha, in line 98, it should be noted that E484K also arose in this lineage, which can confound the results: <https://www.ncbi.nlm.nih.gov/pmc/articles/PMC8820839/>

Also, no positive signal, is a poor indicator of mutations and should be noted, due to the possibility of drop out in PCR if target abundance is low or other mutations preclude the identification of the interrogated mutation.

Line 100 how were the variants defined?

Line 111 it would be clearer to say that there was no significant difference between Ct's of these samples, then to list values.

117 difference due to PCR efficiency? No need to run a significance test on this, unless proving a point or testing something.

Line 136 how was the variant defined as beta if the K417N mutation wasn't present? Should be discussed. How are you defining Beta-positive. This should at least be discussed in the NGS methods section.

Discussion

The authors mention that in BA.1 the mutation rate of K417N is 57%

In lines 193-4 the authors say "reflex" use, and thus preventing the "default" and what they are trying to say is unclear to me.

On line 197 authors incorrectly assert that BA.1 is equivalent to B.1.1.529. where BA.1 is a subvariant of B.1.1.529 originally named B.1.1.529.1

Methods

Do the variant assays have a LOD? Should be discussed

Line 239, these dates should be in the introduction.

I only see 73 sequences in gisaid from your institution and they were uploaded by people not involved in this study. Is there a way for you to confirm that they're uploaded? Also is there a specific tag that you can reference so people can find the sequences if so? Otherwise, another way of sharing the sequences would be needed to validate some of the K417N dropouts mentioned.

Reviewer #1 (Comments for the Author):

VOC have changed quite a lot over time, so this paper should reflect the most recent status, which is that the US CDC only considers Omicron and its subvariants as VOC (<https://www.cdc.gov/coronavirus/2019-ncov/variants/variant-classifications.html>). Whereas WHO states that Omicron is the only currently circulating VOC and Alpha, Beta, Gamma, and Delta are the previously circulating VOC (<https://www.who.int/activities/tracking-SARS-CoV-2-variants>).

Answer: I very grateful to the reviewer for this and the other suggestions/comments listed below. Regarding this important issue, I modified some sentences in the Introduction, page 4, lines 61 to 69 to comply with the changes of VOCs over time. See also the Abstract (page 2, line 40) for consistency.

While the Pango lineages are more specific for mutational patterns, in this context the WHO designated VOC names of Alpha, Beta, etc. should be used when talking about the virus mutations, if no further breakdown of lineage is being discussed. Lines 68-72.

Answer: As suggested, I modified some sentences in the Introduction (pages 4/5, lines 61 to 65 and lines 74 to 79) to make clear the distinction between the two designations.

Around lines 75, the current state of genotyping methods in the field of SARS-CoV2 variants should be discussed in the introduction for proper context of the reader. Ref 15 is a good start.

Answer: As suggested, I added some sentences about the currently available PCR assays in the Introduction (page 5, lines 80 to 82) to provide a proper context to the reader.

When you use d69-70 and N501Y to identify Alpha, in line 98, it should be noted that E484K also arose in this lineage, which can confound the results (<https://www.ncbi.nlm.nih.gov/pmc/articles/PMC8820839/>)

Answer: To satisfy the reviewer's comment, I modified the sentences (see Results, page 6, lines 109 to 112 and lines 118 to 121) to add clarity about the detection of mutations with the Seegene assays.

Also, no positive signal, is a poor indicator of mutations and should be noted, due to the possibility of drop out in PCR if target abundance is low or other mutations preclude the identification of the interrogated mutation.

Answer: To consider the reviewer's comment, I added some sentences (see Results, page 7, lines 136 to 138) about the possible interpretations of negative PCR results.

Line 100 how were the variants defined?

Answer: Regarding how the variants were defined, I added a specification at that point (page 5, line 103) and, consistently, I enriched the NGS methods section with many details (see also the answer to the specific comment below).

Line 111 it would be clearer to say that there was no significant difference between Cts of these samples, then to list values.

Answer: As suggested, I modified the sentence for clarity. See page 6, lines 121 to 123.

Line 117 difference due to PCR efficiency? No need to run a significance test on this, unless proving a point or testing something.

Answer: As suggested, I omitted statistical analysis results. See page 6, lines 126 to 127.

Line 136 how was the variant defined as beta if the K417N mutation was not present? Should be discussed. How are you defining Beta-positive. This should at least be discussed in the NGS methods section.

Answer: As mentioned above, I enlarged the NGS methods section to detail how the variants were defined as well as I moved the relative paragraph before the paragraph dealing with SARS-CoV-2 variant detection testing. See pages 11/12, lines 257 to 282, and a new Table (Table S1 in the supplemental material), which summarizes the additional information.

Discussion. The authors mention that in BA.1 the mutation rate of K417N is 57%.

Answer: I modified the sentences at lines 197 to 203 to clarify the relevant issue.

In lines 193-4 the authors say” “reflex” use, and thus preventing the “default”” and what they are trying to say is unclear to me.

Answer: I modified the sentence (page 9, lines 205 to 206) for clarity.

Line 197 authors incorrectly assert that BA.1 is equivalent to B.1.1.529. where BA.1 is a subvariant of B.1.1.529 originally named B.1.1.529.1.

Answer: I apologize for the typo. I modified the designation appropriately. See page 9, line 208.

Methods. Do the variant assays have a LOD? Should be discussed.

Answer: As required, I added information about the assay’s LOD. See page 13, lines 301 to 304.

Line 239, these dates should be in the introduction.

Answer: As required, dates were also mentioned in the Introduction (page 5, lines 94 to 97).

I only see 73 sequences in GISAID from your institution and they were uploaded by people not involved in this study. Is there a way for you to confirm that they are uploaded? Also is there a specific tag that you can reference so people can find the sequences if so? Otherwise, another way of sharing the sequences would be needed to validate some of the K417N dropouts mentioned.

Answer: I apologize for not checking before that the sequences were all correctly uploaded on the GISAID database. Now, whole genome sequences are available on the database, whereas S-gene sequences are available on the Sequence Read Archive (SRA) repository under accession number PRJNA880628. See page 12, lines 274 to 275.

Reviewer #2 (Comments for the Author):

Authors need to address the sensitivity, specificity, positive predictive values of the assays.

Answer: As specified in the Methods (Statistical Analysis), regarding the clinical performance of the Seegene assays, we calculated the positive percent agreement (i.e., sensitivity) for each of the Seegene assays evaluated in the study using the NGS as the reference method, because only SARS-CoV-2 positive samples were included in the study. Additionally, we included data about the sensitivity, specificity, positive predictive value, and negative predictive value, which were calculated per single variant detection (i.e., Alpha, Beta, Gamma, etc.), as mentioned in the text and detailed in a new supplemental Table (Table S2). See page 14, lines 319 to 321, and Table S2.

October 24, 2022

Prof. Maurizio Sanguinetti
Fondazione Policlinico Universitario
Microbiology
L.go A. Gemelli 8
Rome, RM 168
Italy

Re: Spectrum02922-22R1 (Two-Period Study Results from a Large Italian Hospital Laboratory Attesting SARS-CoV-2 Variant Detection PCR Assays' Evolution)

Dear Prof. Maurizio Sanguinetti:

Your manuscript has been accepted, and I am forwarding it to the ASM Journals Department for publication. You will be notified when your proofs are ready to be viewed.

Sincerely,

Heba Mostafa
Editor, Microbiology Spectrum
